# Potential of Water Hyacinth Infestation on Lake Tana, Ethiopia: A Prediction Using a GIS-Based Multi-Criteria Technique

**Minychl G. Dersseh** [1,*]**, Aron A. Kibret** [2]**, Seifu A. Tilahun** [1]**, Abeyou W. Worqlul** [3]**,**
**Mamaru A. Moges** [1]**, Dessalegn C. Dagnew** [4] **, Wubneh B. Abebe** [5] **and Assefa M. Melesse** [6]

[1] Faculty of Civil and Water Resources Engineering, Bahir Dar Institute of Technology, Bahir Dar University, Bahir Dar P.O. Box 26, Ethiopia; satadm86@gmail.com (S.A.T.); mamarumoges@gmail.com (M.A.M.)
[2] Amhara Bureau of Water Resources, Irrigation and Energy, Bahir Dar P.O. Box 88, Ethiopia; atekaaron@gmail.com
[3] Texas A&M AgriLife Research, Temple, TX 76502, USA; abeyou_wale@yahoo.com
[4] Institute of Disaster Risk Management and Food Security Studies, Bahir Dar University, Bahir Dar P.O. Box 26, Ethiopia; cdessalegn@yahoo.com
[5] Amhara Design and Supervision Works Enterprise (ADSWE), Bahir Dar P.O. Box 1921, Ethiopia; wubnehb@yahoo.com
[6] Department of Earth & Environment, Florida International University, Miami, FL 33199, USA; melessea@fiu.edu
* Correspondence: minychl2009@gmail.com; Tel.: +251-918-702-284

**Abstract:** Water hyacinth is a well-known invasive weed in lakes across the world and harms the aquatic environment. Since 2011, the weed has invaded Lake Tana substantially posing a challenge to the ecosystem services of the lake. The major factors which affect the growth of the weed are phosphorus, nitrogen, temperature, pH, salinity, and lake depth. Understanding and investigating the hotspot areas is vital to predict the areas for proper planning of interventions. The main objective of this study is therefore to predict the hotspot areas of the water hyacinth over the surface of the lake using the geographical information system (GIS)-based multi-criteria evaluation (MCE) technique. The main parameters used in the multi-criteria analysis were total phosphorus ($>0.08$ mg L$^{-1}$), total nitrogen ($>1.1$ mg L$^{-1}$), temperature ($<26.2$ °C), pH ($<8.6$), salinity ($<0.011\%$), and depth ($<6$ m). These parameters were collected from 143 sampling sites on the lake in August, December (2016), and March (2017). Fuzzy overlay spatial analysis was used to overlay the different parameters to obtain the final prediction map of water hyacinth infestation areas. The results indicated that 24,969 ha (8.1%), 21,568.7 ha (7.1%), and 24,036 ha (7.9%) of the lake are susceptible to invasion by the water hyacinth in August, December, and March, respectively. At the maximum historical lake level, 30,728.4 ha will be the potential susceptible area for water hyacinth growth and expansion at the end of the rainy season in August. According to the result of this study, the north and northeastern parts of the lake are highly susceptible for invasion. Hence, water hyacinth management and control plans shall mainly focus on the north and northeastern part of Lake Tana and upstream contributing watersheds.

**Keywords:** water hyacinth; Lake Tana; MCE; fuzzy overlay; source area; Upper Blue Nile

## 1. Introduction

The invasive free-floating aquatic plants have been the main challenges of water bodies in the world because they form a thick floating mat on the water surface that tends to reduce sunlight penetration

and the exchange of gases between the water surface and the atmosphere [1–5]. Water hyacinth (*Eichhornia crassipes* [Mart.] Solms), is a perennial, herbaceous, free floating aquatic plant originating in the Amazon Basin, South America [6]. It can grow in stagnant and flowing water bodies [7]. The species was discovered by a German naturalist, C. von Martius, who was studying the flora of Brazil in 1823 [8]. Since the late 1800s, the plant has spread into large areas of the world [8,9] and then, starting from the Amazon basin reached many tropical and sub-tropical countries of Latin America, the Caribbean, Africa, Southeast Asia, and the Pacific [10,11]. As stated by [11], remarkable freshwater bodies, swamps, and wetland areas of Africa and the Middle East have been invaded by this invasive weed. The reason for the wide spread and growth of the weed in these areas might be due to eutrophication caused by a high level of nutrients. Factors that can create favorable conditions for this growth can be salinity, pH, temperature, sunlight shading, disturbance, and reproduction systems of the invasive weed [12].

In Ethiopia, water hyacinth was observed and reported for the first time in 1965 in Lake Koka and River Awash, and from there it started to spread to other nearby water bodies [13–15]. Since 2011, water hyacinth has invaded Lake Tana, which is the largest lake in the highlands of Ethiopia [16]. The exact cause and source of the water hyacinth infestation in Lake Tana are not clearly known yet but are probably favored by factors such as sedimentation, extensive fertilizer application in the agricultural parts of the catchment, and pollutants (nutrients) from the surrounding cities (mainly Bahir Dar and Gondar). An estimated sediment load ranging from 15.4 to 60 t ha$^{-1}$ yr$^{-1}$ [17–19] is transported from the catchments to the lake. Different studies have estimated that the sediment deposited in the lakebed is in the range of 12–37 million t yr$^{-1}$ [19–21], and 2.15 million t yr$^{-1}$ of organic matter are deposited in the lake bed [19]. The sediment deposited in the flood plain is estimated to form 82–96% of the total sediment which is transported from the watersheds [20] and comprises 2.57 (±0.17) million t yr$^{-1}$ [19]. According to [19], 1.09 million t yr$^{-1}$ of sediment leave Lake Tana, the rate of sedimentation of Lake Tana is 11.7 ± 0.1 kg m$^{-1}$ yr$^{-1}$, and the trap efficiency of the lake is estimated at about 97%. This sediment transported from the watersheds and deposited in the lake and flood plain is rich in nutrients [19] that can accelerate the eutrophication process of the lake. The water quality of the lake is deteriorating due to effluents from point and non-point sources, i.e., cities and agricultural fields [22,23].

Water hyacinth has a strong impact on the physicochemical components of the water of the invaded ecosystem. Its presence in the form of floating carpets on the surface leads to drastic reductions in temperature, pH, oxygen concentration, and the amount of nutrients in the water column [24,25]. In some cases, it can lead to the total collapse of dissolved oxygen, resulting in the death of the zoocenoses present, in particular of the ichthyofauna [8]. The weed causes serious environmental and socio-economic problems for millions of riparian communities across the world and becomes an additional limiting factor of development. As claimed by [26], the weed has an ability to form a floating dense mat on the water surface which deteriorates the water quality used for irrigation. It also affects fishing and transport for riparian communities [27]. In Ethiopia, the weed causes serious socioeconomic effects [15,28]. As stated by [14], the economic impact of water hyacinth in the invaded areas across Ethiopia was estimated at about US$ 100,000 between 2000 and 2013, for controlling and removing of the weed. The socioeconomic activities of the people whose livelihoods are directly or indirectly contingent on the ecosystem services of Lake Tana, are affected by the water hyacinth invasion. Fishing is becoming painful for the fishermen and the transportation of small boats presents a challenge because of the thick mat formation of the weed. Animal grazing land and agricultural land of the flood plain in the northeastern part are covered by the invasive weed, affecting the agricultural practices [29–31]. Since it is a recent phenomenon in Ethiopia, there are only very few studies regarding the water hyacinth, such as [29–31], which makes it difficult to obtain better understanding of the causes, how it expands, and where it expands in order for the management of the water hyacinth to be knowledge-based.

The reporting of the invaded coverage area is a source of conflict among the stakeholders. According to [29], the invaded area of the lake was 20,000 ha, 50,000 ha, and 34,000 ha in 2012, 2014,

and 2015, respectively, whereas the government offices estimated the coverage to be below 5000 ha in the peak growing season. In a survey done from 4–14 October 2018, the water hyacinth invaded area was estimated as 2279.4 ha of the lake, excluding the infestation area in the flood plain. A scientific approach is necessary to better understand the spatiotemporal variation of the area covered by the water hyacinth to better support management actions.

The prediction of the hotspot areas for water hyacinth infestation could be done using multi-criteria spatial modeling, which is one of the important components of a spatial decision support system (SDSS), which often requires a common scale of values for diverse and dissimilar inputs to create an integrated analysis [32].

In a multi-criteria analysis, the major factors controlling the growth of water hyacinth in water bodies are required which could be categorized as climatic conditions (temperature, humidity, and sunlight shading), water body conditions (salinity of the water, disturbances like flooding, current and wave directions), eutrophication (nitrates, phosphates, and sulphates in dissolved form), pH, and reproduction system (sexual and asexual) of the invasive weed [8,12,33]. The most favorable conditions are stagnant water, shallow water depth (<6 m) [34], surface sediments rich in organic matter, and nutrients such as nitrogen and phosphorous. Water hyacinths highly responded when P is increased from 0.06 mg $L^{-1}$ to 1.06 mg $L^{-1}$, and after this value, the weed did not increase in biomass [35].

This work will contribute to the prediction of the hot spot areas using multiple and suitable factors from the scientific community. It will also help to prepare a preventive approach strategy in the management of water hyacinth, and major factors that dictate the expansion of the weed will be identified. The objectives of this study were therefore to (1) evaluate the threshold values of the major factors for water hyacinth in the context of Lake Tana and (2) to predict the potential hotspot areas of water hyacinth expansion using a geographical information system (GIS)-based multi-criteria analysis. This study will be useful in the implementation of a strategic management and control plan for water hyacinth by focusing on the hotspot areas of the lake.

## 2. Materials and Methods

### 2.1. Study Area

Lake Tana is located in Northwestern Ethiopia, with an average elevation of 1786 meters above mean sea level (m a.s.l.). Geographically, it is located in the range of 10°45′54.1″ N, 36°10′24.9″ E and 12°50′15.9″ N, 38°50′54.48″ E (Figure 1). The lake has a width of about 66 km from east to west and length of 92 km from north to south. The lake covers about 3156 km², with a total drainage area of 15,000 km² [36]. The lake receives 93% the inflow from four major rivers, namely Gilgel Abay, Gumara, Rib, and Megech [37]. In addition to gauged rivers (rivers which have hydrological stations to measure flow data), more than 40 ungauged tributaries (tributaries which do not have hydrological stations to measure flow data) flow into the lake. The major part of the Lake Tana basin is suitable for crop production. According to [20], 55% of the catchment area is used for cultivation, 21% consists of water, 10% is grassland and 1.6% is wetland. The cultivated land and surface runoff have shown an increasing trend between 1973 and 2010 [38]. The basin has an irrigation potential of 112,505 ha [39]. Urbanization of the Lake Tana basin is expanding, but there are no integrated waste management practices [40]. The load of non-point source pollutants is increasing in the Tana basin and in the highlands of the basin [23].

The annual water inflow to the lake was estimated to be 3843 million m³ $yr^{-1}$ from rainfall, 3970 MCM $yr^{-1}$ from gauged rivers, and 2729 MCM $yr^{-1}$ from ungauged rivers [41]. According to [41], the annual evaporation from the lake and annual outflow at the outlet of the Blue Nile were estimated as 5182.5 MCM $yr^{-1}$ and 4714 MCM $yr^{-1}$, respectively. The lake water level fluctuates annually and seasonally with the change of precipitation patterns [36]. According to the Ethiopian Ministry of Water, Irrigation, and Energy (EMoWIE), the maximum weir operating level at Chara-Chara Weir

(outlet of the Blue Nile River) is 1786.35 m a.s.l., and the maximum historical water level at Lake Tana was observed as 1788.2 m a.s.l. The basin is severely affected by erosion and sedimentation which deteriorates the quality of the water bodies in the basin [17,37,42,43]. According to [37], 12–30.50% of the basin has a high erosion potential.

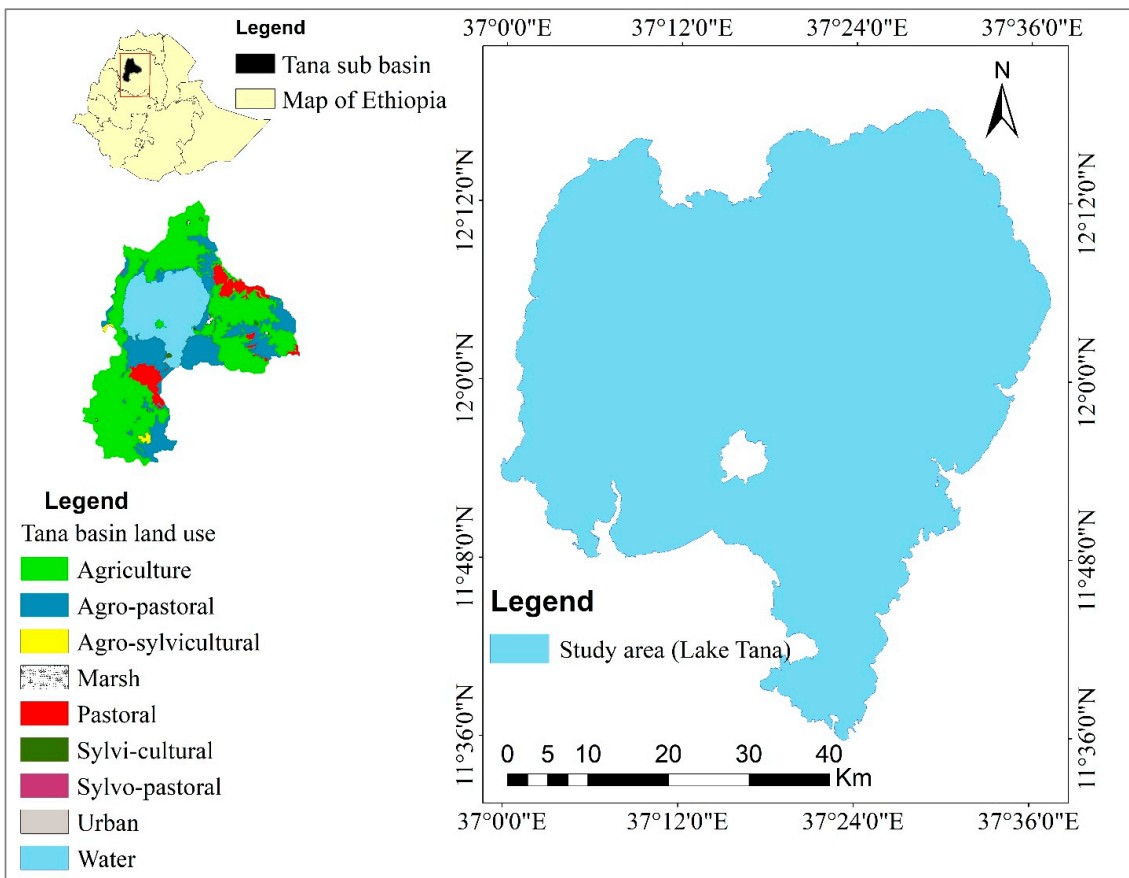

**Figure 1.** Tana basin land use and Lake Tana in the Upper Blue Nile basin (UBN), Northwestern Ethiopia.

*2.2. Dataset*

Water quality data such as total phosphorus (TP), total nitrogen (TN), water surface temperature (T), total dissolved solids (TDS), and pH were collected in August, December (2016), and March (2017) from a total of 143 sampling sites (Figure 2) across the lake at a 5 km interval and 0.5 m depth. The representative sample of each point was then kept in polyethylene bottles and stored at −4 °C until it was analyzed. Total phosphorus (TP) concentrations were determined using PhosVer®3 (Hach Company, Loveland, CO, USA) based on the acid-per-sulfate digestion method in the range of 0.06–3.50 $PO_4^{3-}$ mg P $L^{-1}$. Total nitrogen (TN) concentrations were also determined using the acid-persulfate digestion method in the low range of 0.5–25.0 mg N $L^{-1}$). Digestion was realized at 105 and 150 °C for 30 min, respectively, for TN and TP. The absorption was then measured using Hach product DR.2008 and DR.3900 spectrophotometer at a wavelength of 410 nm and 890 nm for TN and TP, respectively.

The temperature of the water surface was determined in-situ using a WM-32EP EC/pH meter (DKK-TOA Corporation, Tokyo, Japan). Transparency of the water was measured by a Secchi disc of 20 cm in diameter (Abel's Boat maintenance and assembly microenterprise, Bahir Dar, Ethiopia). The maximum depth at which the disc can be seen when lowered into the water is marked and measured.

The reason why we selected the three months was to see temporal variations of the water quality parameters in the rainy and dry seasons due to seasonal surface water fluctuation of the lake. The bathymetry survey conducted by the Tana sub-basin Authority (TaSBo, 2014) was also collected to understand the spatial distribution of the lake depth in August 2014.

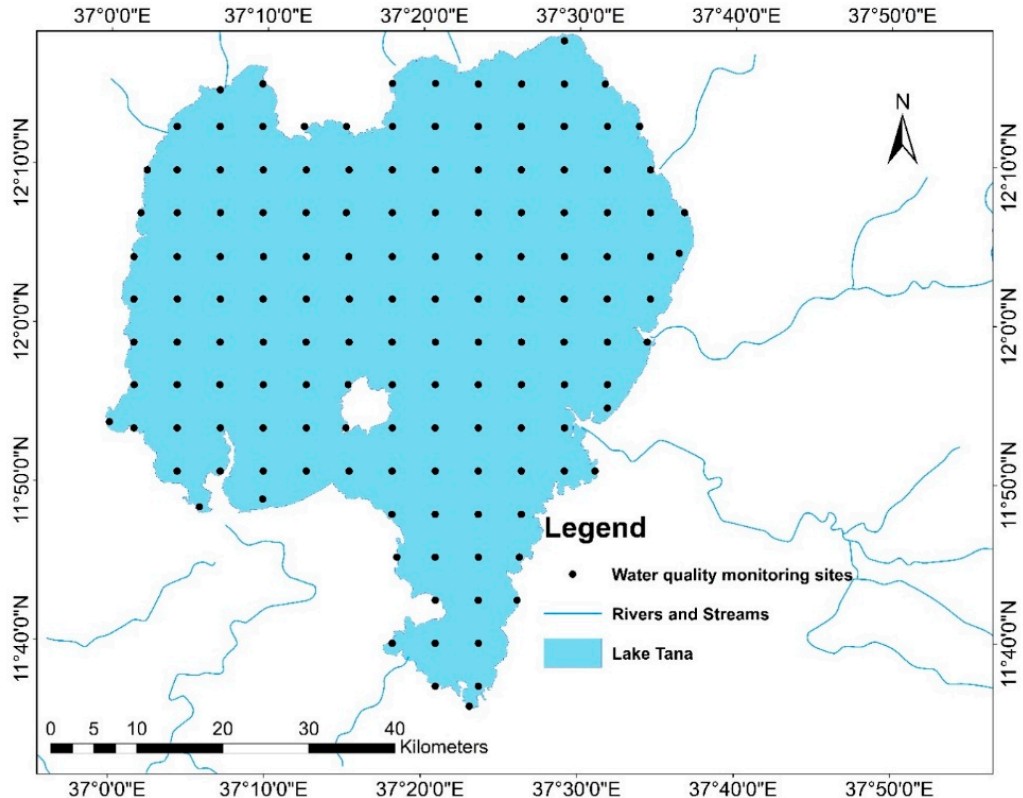

**Figure 2.** Lake water quality data sampling sites (Collected in August, December 2016, and March 2017).

*2.3. Methods*

To achieve the objectives of this study, the application of ArcGIS tools, especially the spatial analyst tool, was very useful. To predict the water hyacinth hotspot invasion area of Lake Tana, three steps were applied. The first step was interpolation and prediction of the spatiotemporal distribution of each major factor using the measured data across the lake. The second step was validation of the threshold values of each major factor by using a map of the invaded area which was prepared by Global Positioning System (GPS) tracking method. Finally, using a fuzzy overlay technique, we predicted and prepared maps of the spatiotemporal hotspot invasion areas of the lake.

2.3.1. Interpolation

The spatiotemporal variability of the water quality data was predicted by ordinary kriging interpolation method, with a spherical semivariogram model using the spatial analyst tool in ArcGIS (ESRI, Redlands, CA, USA). The ordinary kriging method is the most basic method of kriging interpolation, which allows for the application of a statistical model including autocorrelation, rather than applying a deterministic approach by using inverse distance weight (IDW) [44]. Using these interpolated values, raster layers for each parameter were developed, which show the spatial variability of water hyacinth potential infestation areas. Using the maximum historical lake level (1788.2 m a.s.l), the probable flood area of the basin which is connected to the lake water was estimated and mapped using a 30 m resolution digital elevation model (DEM).

2.3.2. The Threshold Value of the Major Factors

The major factors which determine the growth and expansion of water hyacinth are total phosphorus (TP), total nitrogen (TN), water surface temperature (T), salinity, pH, and water depth [12]. The values of water surface temperature, TP, TN, salinity, pH, and water depth which are suitable for water hyacinth growth were determined from previous studies [9,34,45–50]. Some threshold values

were taken from previous research and most of them were fixed after critical evaluation and monitoring of measured parameters and their values in the current invaded area of the lake. The map of the current invaded area was developed by GPS tracking from 4–14 October 2018 when the growth of the weed was highest. Those parameters will help us to identify the main source areas (hotspots) of the water hyacinth in the water bodies. On the basis of information from the literature [34] and field observation and measurement, the depth of the lake which is suitable for optimum growth of water hyacinth is less than 6 m. Based on the threshold setup, a total of six spatial maps were prepared in each month and overlaid to identify the potential water hyacinth infestation area at different times across Lake Tana.

### 2.3.3. Multicriteria Analysis

The suitability analysis was carried out using GIS-based multi-criteria decision analysis (MCDA) [51–53] with the fuzzy overlay function. Even though the fuzzy overlay is computationally complex, it gives more accurate and consistent results than weighted overlay [32]. Before applying the fuzzy overlay, suitable criteria for optimum water hyacinth growth for each water quality parameter were determined using the literature and the current infestation area survey of the lake. The map of the current water hyacinth covered area was used for validation.

### 2.3.4. Fuzzy Overlay

Suitability analysis deals with continuous factors in nature (soil characteristics, climatic parameters) and with socio-economics linguistic parameters (near, nearer, far away) which are difficult to model using Boolean logic because of their vagueness and imprecision [54]. To make such imprecise, incomplete, and vague information precise, fuzzy (probabilistic) logic is preferable [55–57]. The fuzzy overlay is based on set theory and the transformation defines the possibility of membership to sets [32,58]. According to [32,58], the fuzzy overlay function is better than weighted overlay in suitability analysis. Since fuzzy logic is based on set theory to determine whether a particular location belongs to one or multiple sets, weighting does not make sense, so that increasing the weight of one factor over another cannot increase the possibility of belonging to one set or a combination of multiple sets, as the result, weightings of the criteria are not applicable.

In this study, we applied a membership type of "MSSmall" because it can calculate the membership based on the mean and standard deviation of the input data, where small values have high membership with one and two multipliers of mean and standard deviation, respectively. It can overlay data in a "fuzzy" manner and is useful in identifying the least common denominator for the membership of all the input criteria. Figure 3 shows the flow chart of the method used.

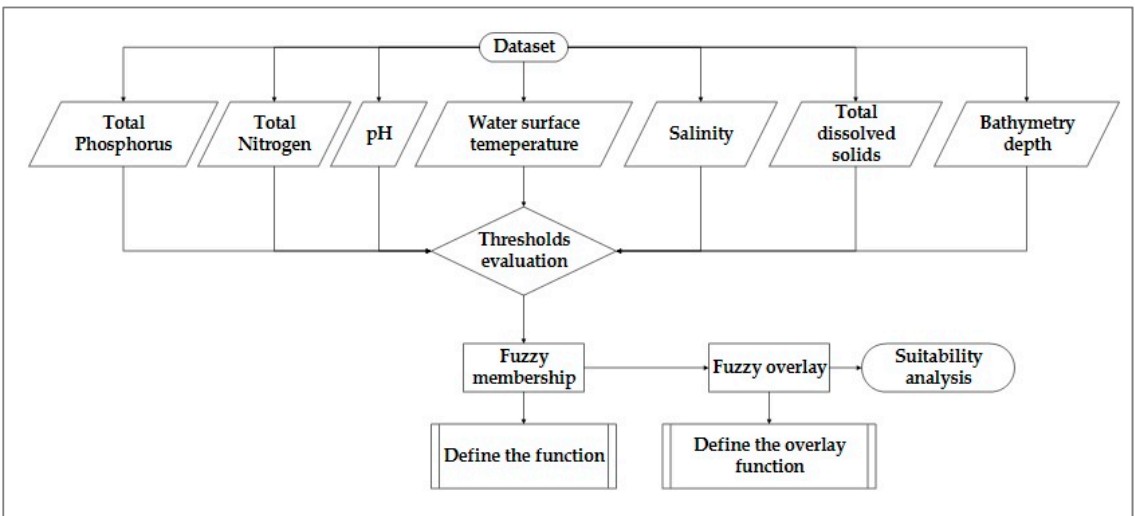

**Figure 3.** Flow chart for multi-criteria spatial model using fuzzy overlay developed to identify the potential water hyacinth infestation area.

## 3. Results and Discussion

### 3.1. Threshold Water Quality Parameters

In this study, six water quality parameters were considered in the multi-criteria analysis for the prediction of hotspot locations and areas in the rainy (August) and dry seasons (December and March). The spatiotemporal distribution of selected parameters has been interpolated and presented below (Figure 4).

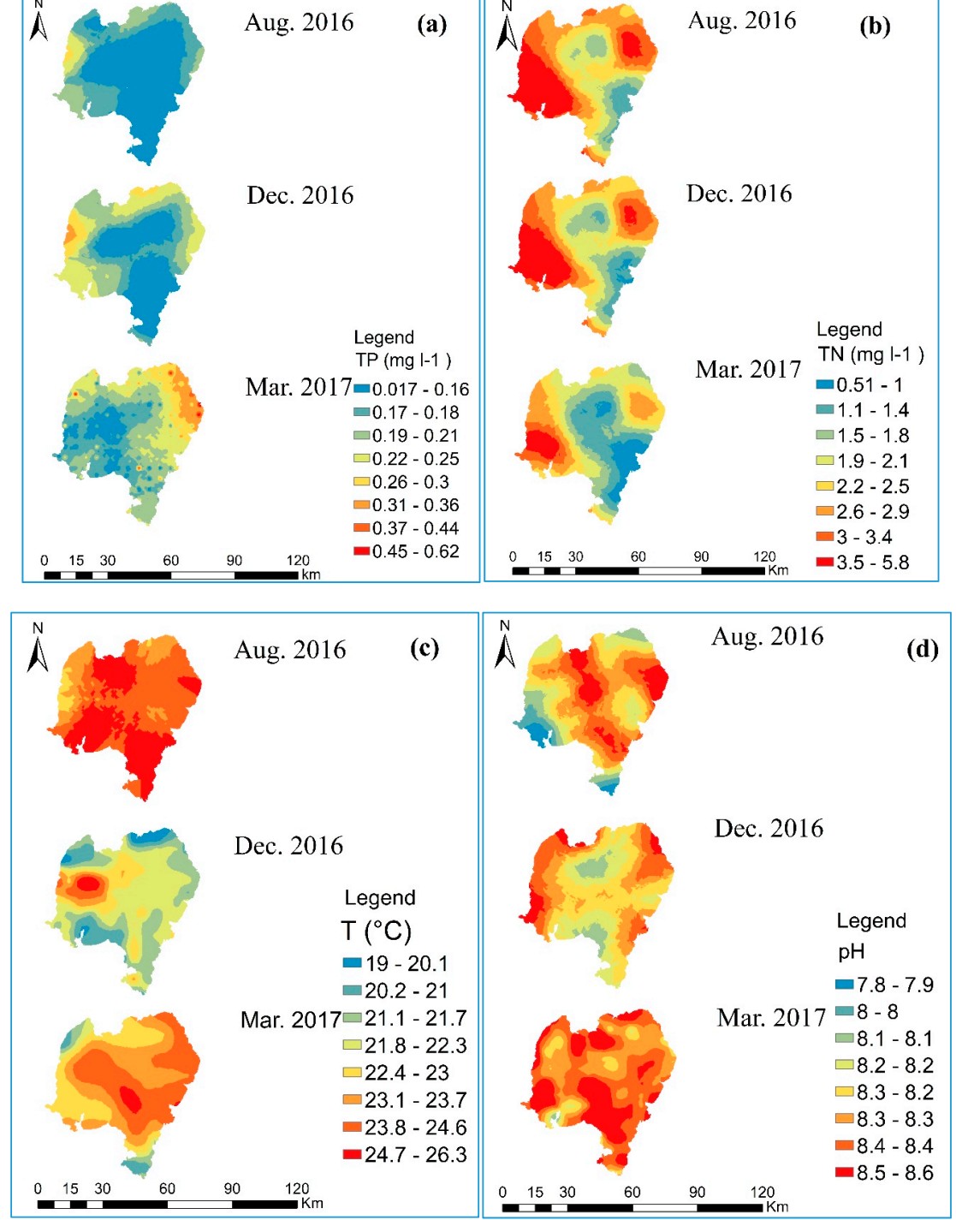

**Figure 4.** *Cont.*

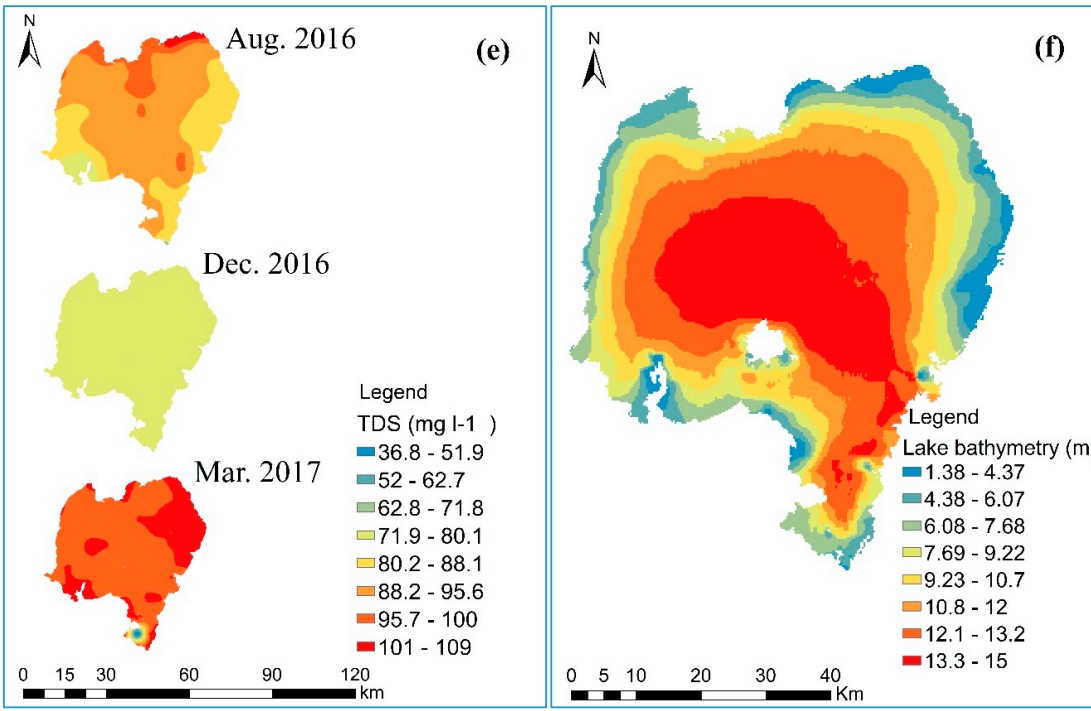

**Figure 4.** Spatial variability of total phosphorus (**a**), total nitrogen (**b**), temperature (**c**), pH (**d**), total dissolved solids (**e**), and lake bathymetry (**f**) over Lake Tana (August and December 2016 and March 2017, except depth data).

In Figure 4, the highest values of the total phosphorus (TP) concentration were observed in the dry season (December and March) and the lowest in the rainy season (August) (0.14 mg L$^{-1}$ in August, 0.18 mg L$^{-1}$ in December, and 0.21 mg L$^{-1}$ in March). Phosphorus flowing from the catchment to the lake through runoff might be in particulate and dissolved forms. The particulate form can be re-dissolved and re-suspended by wind and waves [59]. Lakes with a low flushing rate have high retention [60]. Based on the results of [41], we have calculated the flushing rate of the lake and found it to be 18.4% per year of the net inflow, which is very low, and the residence time was estimated to be about 5.4 years. The retained particulate phosphorus can be re-suspended, and its re-suspension is higher in shallow lakes due to wind-induced remixing events [61–63]. The reason behind the maximum value of phosphorus in the dry season in Lake Tana might be the low flushing rate/high residence time, high retention capacity, and wind-induced re-suspension of the particulate phosphorus in the dry months when the temperature increases.

The highest values of total nitrogen (TN) concentration were observed in the rainy season (August) and the lowest in the dry season (in December and March) (2.7 mg L$^{-1}$ in August, 2.6 mg L$^{-1}$ in December, and 1.9 mg L$^{-1}$ in March). This might be due to high nitrogen fertilizer application in the basin and its transport to the lake through runoff by tributary rivers. The agricultural office report indicates that the diammonium phosphate (DAP) and urea fertilizer application for rain-fed and irrigation agriculture is increasing in the highland areas of the basin. The spatial and temporal variability of the pH value of the lake was not substantial and ranged between 7.8 and 8.6, which indicates that the lake is productive or eutrophic in its state and suitable for water hyacinth and other aquatic plants. According to the results (Table 1), the temporal variability of the temperature is not, as such, substantial, but there is some spatial variability of the temperature of the lake. However, the temperature is suitable for the growth and expansion of aquatic plants, including water hyacinth. The spatiotemporal values of the TDS were found in the range of 63.5–108.7 mg L$^{-1}$ in August, 72.1–77.7 mg L$^{-1}$ in December, and 36.8–107.2 mg L$^{-1}$ in March, which are low values that suitable for the growth of water hyacinth [64]. The range of the TDS values in December was small, which needs further investigation. Substantial shallow depth of the lake is found on the northeast shore which is currently

invaded by water hyacinth. The statistics of the temporal values of the five parameters are summarized and presented in Table 1.

**Table 1.** The temporal values of selected physicochemical water quality parameters.

| S/No | Parameters | Statistical Parameters | Monthly Values | | |
|------|------------|------------------------|--------|----------|-------|
| | | | August | December | March |
| 1 | TP (mg L$^{-1}$) | Max | 0.30 | 0.40 | 0.60 |
| | | Min | 0.05 | 0.07 | 0.02 |
| | | Mean | 0.14 | 0.18 | 0.21 |
| | | St. deviation | 0.04 | 0.05 | 0.05 |
| 2 | TN (mg L$^{-1}$) | Max | 5.7 | 5.6 | 4 |
| | | Min | 1.1 | 0.90 | 0.50 |
| | | Mean | 2.7 | 2.6 | 1.9 |
| | | St. deviation | 0.98 | 0.98 | 0.75 |
| 3 | pH | Max | 8.5 | 8.4 | 8.6 |
| | | Min | 7.8 | 8 | 8 |
| | | Mean | 8.2 | 8.2 | 8.3 |
| | | St. deviation | 0.12 | 0.08 | 0.079 |
| 4 | Temperature (°C) | Max | 26 | 26 | 25 |
| | | Min | 22 | 19 | 20 |
| | | Mean | 24.4 | 22 | 23.3 |
| | | St. deviation | 0.75 | 0.98 | 0.86 |
| 5 | Salinity (%) | Max | 0.011 | 0.008 | 0.011 |
| | | Min | 0.005 | 0.007 | 0.003 |
| | | Mean | 0.009 | 0.007 | 0.010 |
| | | St. deviation | 0.0005 | 0.0002 | 0.0004 |

The minimum and maximum values of each parameter were considered parameter to set the threshold values for optimum conditions of water hyacinth growth (Table 1). Setting the threshold values was done in two ways. The first technique was taking optimum values from the literature and the second was validating the literature values with the current condition of water hyacinth growth on Lake Tana using spatiotemporal, measured parametric values. It was found that except for depth, most of the conditions found in the literature could not be fitted with the Lake Tana conditions. To solve this dissimilarity, we decided to validate the threshold values of Lake Tana condition by critically evaluating the linkage between the current invaded area and the measured spatial values of the parameters. The validation was carried out by using the map showing the current status of water hyacinth invasion, which was prepared through tracking the invaded areas of the lake using handheld GPS and discussions held with the local community.

Figure 5 indicates that almost all pH, salinity, and water surface temperature conditions of Lake Tana are appropriate for the growth of water hyacinth. The lake is also highly suitable in terms of nutrients (TP and TN) for the growth of water hyacinth because the spatially measured values were found in optimum ranges, as supported by the literature (Table 2). The suitable depth for water hyacinth on Lake Tana was found on the northeast shore of the lake, except small areas in the southwest corridor. The threshold values of each selected parameter from the literature and the Lake Tana condition are reported in Table 2.

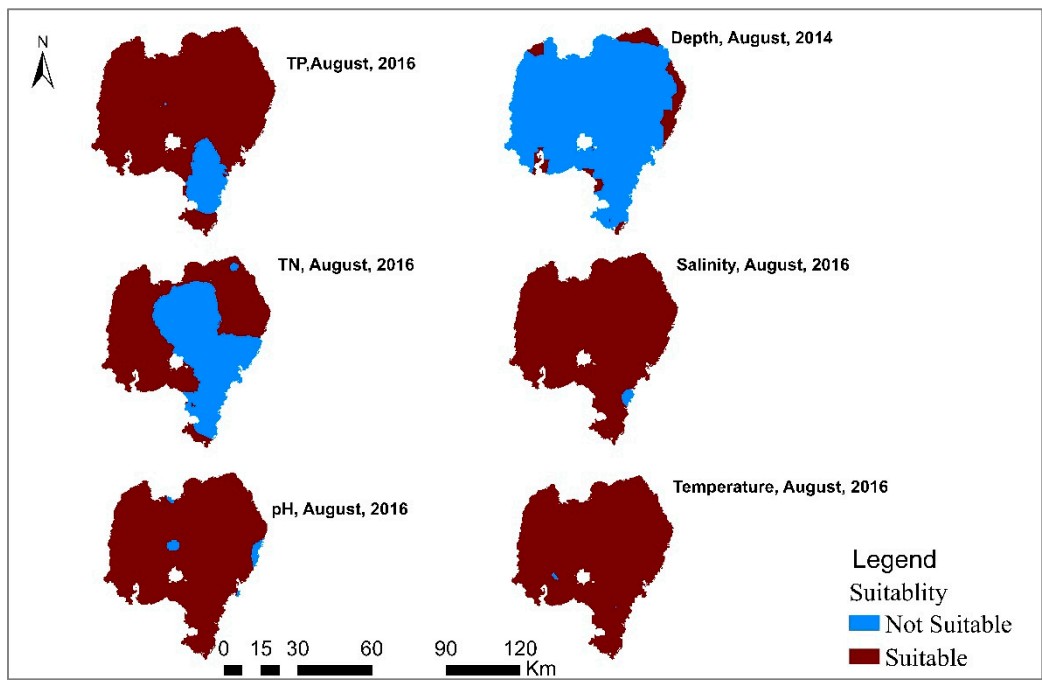

**Figure 5.** Suitable area of Lake Tana in terms of major factors for water hyacinth growth in August.

**Table 2.** Threshold values of water quality parameters in comparison with previous studies.

| S/N | Parameters | Range | References | In This Study |
|---|---|---|---|---|
| 1 | TP | 0.02–0.10 mg L$^{-1}$<br>1.66–3 mg L$^{-1}$ | [50]<br>[12] | >0.08 mg L$^{-1}$ |
| 2 | TN | 0.50–1 mg L$^{-1}$<br>0.05–1 mg L$^{-1}$<br>5.50–20 mg L$^{-1}$ | [45]<br>[50]<br>[12] | >1.1 mg L$^{-1}$ |
| 3 | T | 8–40 °C<br>12–38 °C<br>10–40 °C<br>28–30 °C | [48]<br>[45]<br>[50]<br>[12] | <28 |
| 4 | pH | 4.0–10.0<br>6.0–8.0<br>6.5–8.5 | [48]<br>[47]<br>[12] | <8.6 |
| 5 | Salinity | <0.0005%<br>0.13–0.19%<br><0.2% | [46]<br>[49]<br>[12] | <0.011% |
| 6 | Depth | <6 m | [34] | <6 m |

*3.2. Hotspot Area Prediction of Water Hyacinth Area Using MCDA*

Six physical and water quality parameters such as depth, TP, TN, pH, TDS, and T were considered to establish layers of the MCDA. The threshold values established in this study were: depth < 6 m, total phosphorus > 0.08 mg L$^{-1}$, total nitrogen > 1.10 mg L$^{-1}$, a temperature range of 23–27 °C, pH of 8–8.60 and salinity < 0.011%. After the determination of the threshold values for each parameter, the spatial maps were prepared for each parameter using ArcGIS (ESRI, Redlands, CA, USA), Spatial Analyst tool, and Reclassify tool. Through fuzzy overlay analysis, the maximum invasion area of the lake was predicted (Figure 6).

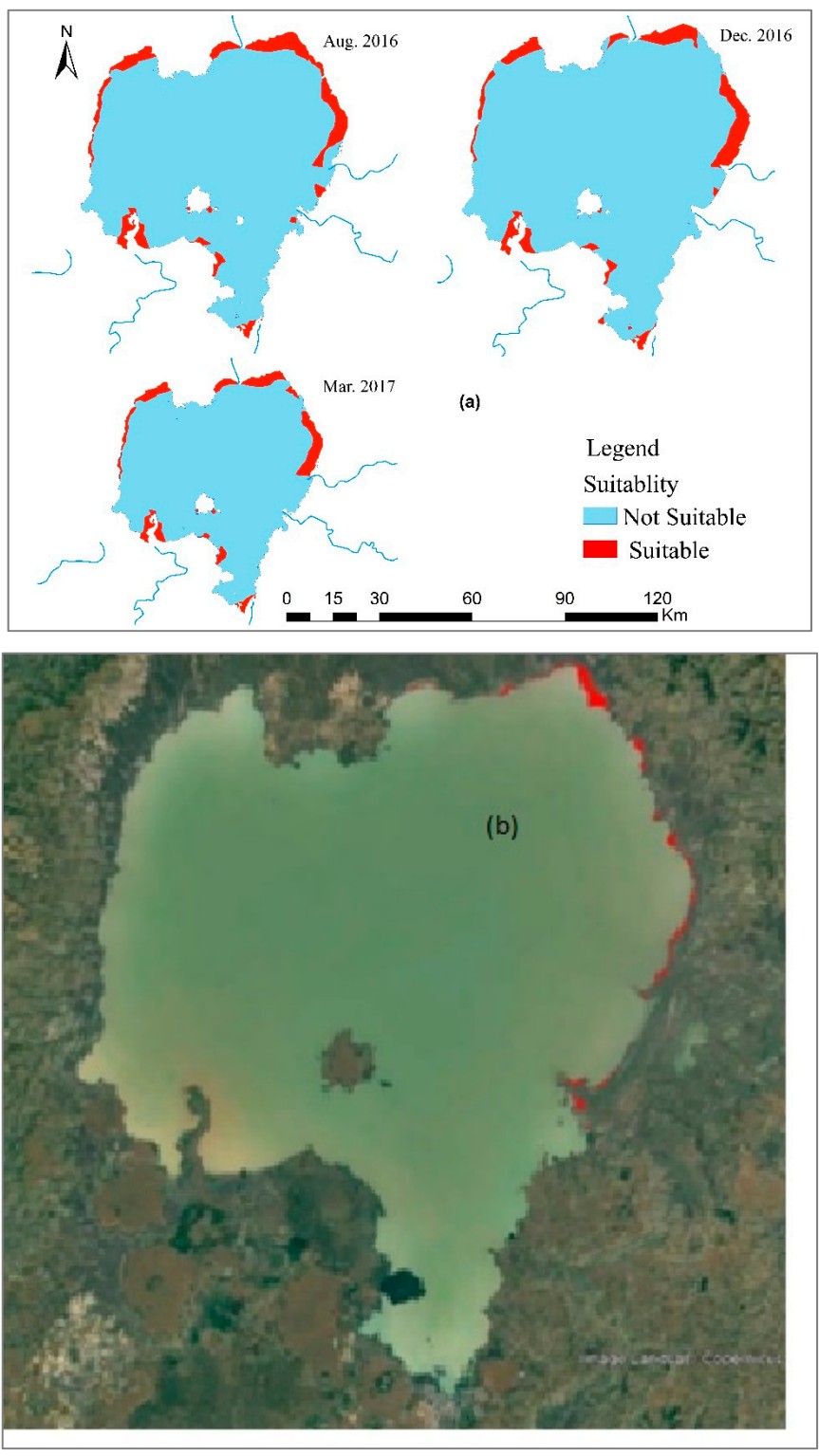

**Figure 6.** Predicted suitable area using multi criteria analysis for water hyacinth growth in different months (**a**). Map of Lake Tana and its invaded area by water hyacinth (red color) from 6–10 October 2018 (prepared by GPS tracking and overlaying on Google Earth) (**b**).

Currently, the hotspot areas which have been invaded by the invasive species are located on the northeastern shore of the lake (6–10 October 2018, shown in Figure 6b). In the rainy season (August), the western part is also susceptible to be invaded by the weed, but so far, there was no observation of an invasion. According to seasonal lake level fluctuations, the surface water

area of Lake Tana ranges from 2966 km$^2$ to 3100 km$^2$ [65]. On the basis of the data from Amhara Design and Supervision Works Enterprise, the surface water area was in the range of 3035.3 km$^2$ (dry season)–3091.32 km$^2$ (end of the rainy season). The magnitude of the invaded area has been affected by the seasonal surface water fluctuation of the lake [66]. According to the Ethiopian Ministry of Water, Irrigation, and Electricity (EMoWIE), the maximum weir operating level at Chara-Chara (outlet of the Blue Nile River) is 1786.35 m a.s.l. and the maximum historical Lake Tana level is 1788.2 m a.s.l. At the maximum probable lake level (1788.2 m a.s.l.), a 5759 ha flood plain which is susceptible to be invaded by water hyacinth may be covered by flooding water due to the back flow of the lake water. The reason why the northeastern shore of the lake is susceptible to invasion might be due to the large flood plain which has a great potential to sink nutrients [67] and the substantial amount of sediment deposition from the major rivers (Gumara) [68], which causes flooding in this area and makes it suitable for water hyacinth growth and expansion. The overall suitable and non-suitable areas are summarized in Table 3.

**Table 3.** Summary of the predicted suitable and not-suitable areas for water hyacinth growth.

| Month | Area | Suitable | Not-Suitable | Total |
|---|---|---|---|---|
| August | Area (ha) | 24,969 * | 284,163 | 309,132 |
| | % | 8.1 | 91.9 | - |
| December | Area (ha) | 21,568.7 | 281,961.8 | 303,530.5 |
| | % | 7.1 | 92.9 | - |
| March | Area (ha) | 24,036 | 279,494.5 | 303,530.5 |
| | % | 7.9 | 92.1 | - |

* The suitable area does not include the flood-susceptible area (5759.4 ha) due to the probable maximum historical lake level.

The area suitable for infestation was found to be larger in August than in March and December (Table 3). This result is supported by the actual conditions of the infestation of the weed (Figure 4). In the last five or six years, the infestation has worsened at the end of the rainy season in the northeast corridor and has spread a little further southeast of the lake. In the dry season, the infestation is limited to the mouth of the perennial rivers and the littoral portion of the lake. The main reasons for this variation might be the increase of the lake's water level and the occurrence of flooding as the result of the Chara-Chara weir at the outlet of the Blue Nile River. At the end of the rainy season, the large flood plain is flooded and is suitable for the growth of water hyacinth. Including the estimated flood area, the potential susceptible area for water hyacinth growth and expansion will be at its maximum in August, with a total area of 30,728.4 ha (Figures 6 and 7).

Although the southwest and northwest corridors of the lake are suitable for water hyacinth growth on the basis of the multi-criteria evaluation, the area is still free from the floating invasive weed, as shown in Figure 6b. Although further data collection and investigation is needed, the reason might be due to the wind and wind-induced wave direction. The floating weed moves into bays and quiescent parts of the Lake water surface due to the wind and wind-induced wave direction [69]. Based on personal observations during field work and direct communication with local people, it seems that the direction of the wind during the daytime is towards the northeast and the waves move in the same direction. At the same time, it is observed and verified that the growth of weeds is stable in the stagnant water bodies and in the littoral areas of the lake.

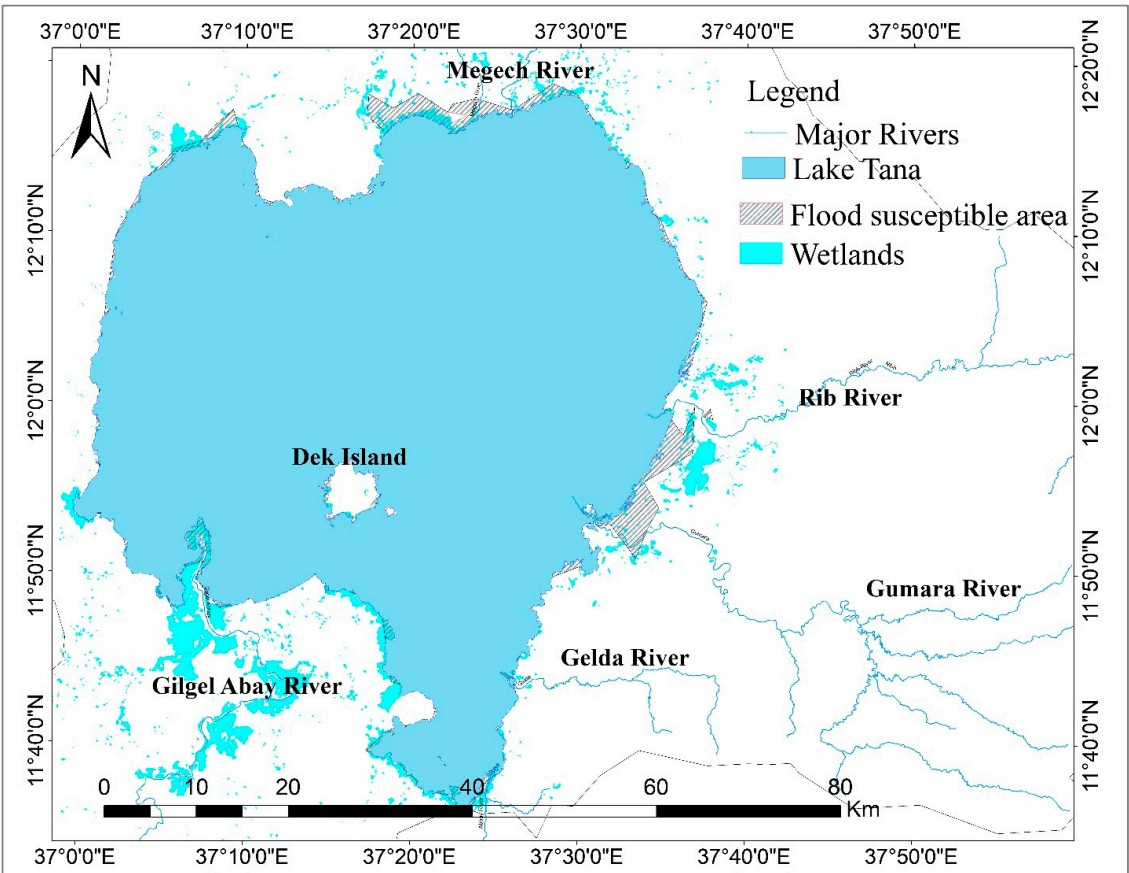

**Figure 7.** Maximum probable flooding area in the Tana basin at the maximum historical lake level (1788.2 m a.s.l.).

## 4. Conclusions

The major factors applied in the multi-criteria analysis to predict the hotspot areas of water hyacinth growth on Lake Tana were total phosphorus, total nitrogen, temperature, pH, salinity, and depth. According to the results, a substantial area of the lake is susceptible to water hyacinth invasion, especially the northeastern part of the lake. This might be due to the large flood plain in the northeastern part, the wind direction towards it, and the shallow lake depth in this area. The maximum invasion area of the lake is 30,728.4 ha (10%) in August at the maximum lake water level. Although the results match the existing conditions of the water hyacinth infestation in the lake, it currently only affects the northeastern portion of the lake, as described in Figure 6b. This might be due to the wind direction and wind-induced wave direction that moves the weed towards the east. The weed needs stagnant water, which is typical for the northeastern parts of the lake due to their connection with the large flood plain in this area. The impact of the wind direction and wind-induced wave current has to be investigated critically in future works. This study will help to implement control and management strategies for the stakeholders and the concerned governmental bodies.

**Author Contributions:** M.G.D. contributed in computation of the overall GIS work, data analysis, writing the manuscript and improving the manuscript based on the comments and suggestions of the coauthors. A.A.K. contributed in the data collection and laboratory work. A.M.M. contributed in giving comments and suggestions and improving the writing up and result interpretation. S.A.T. contributed in shaping the objectives, the methods, and in giving comments on the overall work. A.W.W. helped in the GIS part and in improving the quality of images. M.A.M. contributed in shaping the overall contents, English, structure of the paper, and in giving significant comments. D.C.D. contributed in editing, improving the English, and in giving significant comments for the paper. W.B.A. contributed in giving comments and suggestions in the GIS part of this work.

**Funding:** This research was funded by Bahir Dar University Ethiopia and International Water Management Institute (IWMI) as part of Feed the Future Innovation Lab for Small Scale Irrigation (AID-OAA-A-13-0005),

the Feed the Future Innovation Lab for Sustainable Intensification (AID-OAA-L-14-00006) through the Sustainably Intensified Production Systems Impact on Nutrition (SIPSIN) and Blue Nile Water Institute for data collection and by the Amhara Environment, Forest, and Wild-life Protection and Development Authority (AEFWPDA) for GPS mapping.

**Acknowledgments:** The data were collected partly by Bahir Dar University Ethiopia and International Water Management Institute (IWMI) as part of the Feed the Future Innovation Lab for Sustainable Intensification (AID-OAA-L-14-00006) through the Sustainably Intensified Production Systems Impact on Nutrition (SIPSIN) and partly by Blue Nile Water Institute. We thank the Tana Sub Basin Office (TaSBO) and Amhara Design and Supervision Works Enterprise (ADWE) for giving us additional secondary data. Special thanks are also expressed to the Amhara Environment, Forest, and Wild-life Protection and Development Authority (AEFWPDA) to initiate the study and support funding for the GPS mapping.

**Conflicts of Interest:** The authors declare no conflict of interest.

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
