# Peer review of "Potential of Water Hyacinth Infestation on Lake Tana, Ethiopia: A Prediction Using a GIS-Based Multi-Criteria Technique"

_water, doi:10.3390/w11091921_

Round 1

Reviewer 1 Report

for detailed comments and suggestions see attached file

Reviewer 2 Report

I am glad to review the paper intitled Spatiotemporal Prediction of Potential Water Hyacinth Infestation Area using a GIS-based Multi-Criteria Evaluation Technique for Lake Tana, Ethiopia. Such an important topic as the infestation of water hyacinth. The topic is relevant to be explored because of the effect the vegetation has in the water quality and surroundings. The research was conducted in the Lake Tana, in 143 sites and in 3 different periods of the year and compared their data with the literature to make maps of possible infestation of water hyacinth.

Therefore, below you will find my contributions to the paper.

The title Spatiotemporal Prediction of Potential Water Hyacinth Infestation Area using a GIS-based Multi-Criteria Evaluation Technique for Lake Tana, Ethiopia is a bit long and would suggest to short it to

Potential of Water Hyacinth Infestation on Lake Tana, Ethiopia: a prediction using a GIS-based Multi Criteria Technique.

The abstract is well written, but some points must be reviewed. For instance, from line 20 to 22 the researchers must change the order of the sentence. First, the major factors affecting the growth and afterward the necessity of studies. Also, is not clear why lake Tana. The researchers must provide a justification saying why they chose this environment to conduct the work. Moreover, there is no explanation on why to study in three different months. Is it because of the water fluctuation? There is no discussion of the results because the researchers say “According to the result of this study, the north and northeastern part of the lake is highly susceptible to infestation” but they don’t provide an argument on why this may happen.

Please organize the characters. The symbol < sometimes is together, sometimes is separated from the numbers. It is >0.08mg L-1, when it should be > 0.08mg L-1.

Line 30, there is a comma missing “21568.7 ha”

The introduction starts with the definition of water hyacinth and its causes to the environment and society.

In the 46 to 48, there is no citation of the problems. I suggest consulting the paper of Rezania et al 2015 for instance (Perspectives of phytoremediation using water hyacinth for removal of

heavy metals, organic and inorganic pollutants in wastewater)

in the lines 58 and 59, why so many sediment loads numbers? The researchers must provide a justification for why other researchers found so different loads. The same is seen in lines 60 to 61.

Line 65 the researchers say “is rich in that can accelerate …” but, what is rich in? Also, there is no citation.

Line 67, the researchers say that water quality is deteriorating, but why? The sediment load changed the water quality on which point?

From line 68 to 71 the researchers do not show any citation. Why? Is it a personal observation?

The last sentence before the goal does not link to the next paragraph of the objectives. I strongly suggest checking the storyline to make the sentences connect to each other.

Also, strongly suggest to check the languages, there are some small mistakes that need to be taken into consideration

The study area is depicted in the Methods section. However, there is no data about the depth, the seasonality (months of low waters, high waters). How did the researchers sample the water for N, P and other parameters?

I wonder if the literature that the researchers got the data of temperature, phosphate, and others is in the same location, or at least next, of water hyacinth beds. Because the lake is wide and probably deep, therefore, many geochemical processes must have happened.

The results

In December 2016 (figure 3 e) what did happen to the figure? It is so strange that the TDS showed the same value on all the sample sites. I strongly suggest checking the data.

In the figure 3f the researchers did not show the depth in time. Probably will have a discussion afterward, but would be nice to show this in the legend.

The researchers did not say when is the rainy and dry season, therefore, is complicated to distinguish when it is.

There are two figure number 3. Strongly suggest organizing the sequence of the text and figures. Also, the letters do not seem to be the same size and type in lines 234 and 235.

I guess in figure 4, the figure above is a and below is b. But the researchers must indicate.

For the figure 4 would also be very nice to have GPS tracking when there is a bloom of water hyacinth. Does the researcher have this data? It would complete a suitable place showed also in figure 4a.

Is the figure 3 based on literature data? And the figure 4 based on the researcher data?

In figure 3, that should be 4 the red color of the pH is not the same as the others.

If so, it is so strange that the predicted suitable area for water hyacinth infestation in figure 3 was for almost the entire lake with the range from 0.02 mg/L to 1mg/L and the predicted suitable area of the infestation of water hyacinth using the researcher data is so little. Is it because the data of the researcher is less than 0.08 mg/l? If so, how much was it? Because if the prediction with at least 0.02 mg/L gave a map where the suitable area is the entire lake than 0.08 would be the same.

Only in line 258, the authors mention the season.

 The conclusion of the article is much better explained in the abstract than in the text. It would be great to see the % as in the abstract. Also, they do not say about the amount of P and N disposed of in the environment, which drastically will enhance the growth and proliferation of water hyacinth. I found the conclusion a bit vague and should be better explained.

I have read the entire paper and found it important for science. However, there are many changes that the authors must provide, even checking the data. For example, the authors mentioned phosphate and nitrate, but it never appeared again in the results. I wonder if the authors considered Phosphate as TP. I also found a lack of data about wind, which can determine the presence of water hyacinth in the shore of the infestation during certain seasons.

Reviewer 3 Report

Review of the manuscript water-549727, “Spatiotemporal Prediction of Potential Water Hyacinth Infestation Area using a GIS-based Multi-Criteria Evaluation Technique for Lake Tana, Ethiopia” by Dersseh et al. submitted to Water.

This manuscript analyse and predict the potential area of invasion of the water hyacinth in Lake Tana (Ethiopia). The authors identified the major drivers of the water hyacinth growth and their threshold values that were used for the multi-criteria analysis (based on GIS). Then a fuzzy overlay spatial analysis was used to overlay the different parameters to obtain a map of the areas potentially invaded by water hyacinth in different seasons.

The findings of this manuscript are of certain interest, and well located in the item context. The data assembled by the authors have a good potential for an interesting paper in relation to the complexity of the variables analyzed. Some lack of clarity is probably due to the need of a revision of the English language (e.g. grammatical errors, sentence structure…).

The Introduction is almost fluent, and linear up to lead the reader to the formulation of the objectives of the study. Different portions of the paragraph should be revised to better clarify and make more readable the sentences. At the end of the introduction, I think it is important to add one or few sentences about the importance of this work in the international context of the management of water hyacinth as invasive species, and highlight the novelty of this work.

In general, Material and Methods paragraph is quite well organized. Some more details and clarification are needed for “sampling and analysis” in the paragraph 2.2 (dataset). Add the methodology used to determine water quality parameters (see Specific comments). In the paragraph 2.3.2, the methodology and the steps used to identify the main factors and their threshold values should be revised and clarified. Why did you chose these factors? Are they all important to determine water hyacinth extension? The authors well explained the reasons of their choices in the paragraphs 2.3.3 (multicriteria analysis) and 2.3.4 (fuzzy overlay), but they should provide more precise details on the procedure, setting parameters used, etc. to make more understandable the methodology applied. They can explain all these methodological details in Supplementary Material.

The Results and Discussion section is interesting, but it is not always well presented and fluent enough. For example, it is not easy to follow the description of the results and the discussion of the maps reported in figure 3. Moreover, the description of figure 3e is missing. Some parts of the results are more proper M&M, and therefore a re-organization of the section is needed.

The Conclusion is almost proper. It needs only a revision of the English language. I can suggest to stress and discuss the international and multi-layer relevance of their findings, but strictly on the base of their results.

I think that the manuscript can be accepted for the publication after major revision.

See the .pdf file of the manuscript with all the part of the paper that should be revised highlighted in yellow. Below the specific comments on the manuscript.

In general, check the decimals used for the different unit of measure. Check also the references into square brackets, in particular the order of the numbers of the references when you cite more than one references. Check the spaces in all the text. I personally prefer to use “invaded/invasion” or simply “the area covered by..” instead of “infestable/infested”. Please check in all the manuscript and in the title.

Introduction:

Line 50-51: Reformulate the sentence; it is not clear.

Line 57: Which pollutants? Nutrients?

Lines 58-59: The sentence is not clear. What do you mean? The values reported are three different estimates of the sediment loads? Please clarify and reformulate the sentence.

Moreover, What do you mean with sediment loads? The solid transport? Please, specify.

Lines 59-64: as before, the sentences are not clear. Please clarify and use the same units of measure. It is more readable. You can supply a range of values derived by different literature data.

Line 65: a word is missing to complete the sentence. “flood plain is rich in that can accelerate”. Rich in what? Nutrients? Phosphorous?

Line 69: “on Lake Tana”. Do you mean “on Lake Tana water quality”? or quality status? please complete the sentence.

Line 71: “Grazing”. Do you mean “pastures”?

Line 71: “are” instead of “is”

Lines 72-75: reformulate the sentence; it is not fully clear.

Lines 76-82: I suggest to make more readable the paragraph. For example: “The data on the surface covered by water hyacinth is being a ….”; “From literature data the area covered by this invasive species range from 20,000 to 50000 ha... (refs)”.

The vegetative period of water hyacinth covers all the year?

Lines 81-82: You can better reformulate this sentence and merge with the following (lines 83-85). For example: “A scientific approach is necessary to better understand the spatio-temporal variation of the surface covered by the invasive water hyacinth, to better support management actions”.

Line 88: “(SDSS) often requires..”; add “(SDSS) WHICH often requires..”?

Line 95: substitute “quiescent” with “stagnant”

Lines 95-97: Make more readable this part of the sentence. For example. “..., surface sediments rich in organic matter, and nutrients such as nitrogen and phosphorous”.

Material and methods:

Line 107: here and below use the same standard metric measurements: metres above mean sea level (MAMSL) or simply metres above sea level (MASL or m a.s.l.)

Line 114: these percentages are represented in figure 1? Add the reference into the text.

Line 118: substitute “the annual stream flow” with “the annual water inflow”

Line 118: delete “direct” before “rainfall”

Line 126: add “from”; “According to [29], FROM 12 to 30% of the basin..”

Line 132: Delete “spatiotemporal”

Lines 132-134: Add the methodology used to determine water quality parameters. For example, in situ how did you collect water samples (surface water, integrated water along water column…); did you use multiple probe to determine Temperature and pH? Did you determine in lab TP, TN, TDS? With what methodology?

Line 142: substitute “vital” with “very useful”

Line 157: “phosphate and nitrate” or “TP and TN”? There are not correspondence to Table 2. You mention before table 2 than table 1. Rectify in the text.

Line 157: only “nitrate”? I think that also ammonium ion is fundamental as factor for water hyacinth grow, as this ion is more promptly usable by vegetation.

Line 163: You mention before table 2 than table 1. Rectify in the text.

Results and Discussion:

Line 193: The authors should details the six water quality parameters object of the multicriteria analysis. Moreover, as stated before, these six parameters must be detailed in the material and methods section (in the paragraph 2.3.2 for example, which lack of clarity).

Line 194: insert in brackets the months (Aug, Dec., Mar.) included in the rainy and in the dry seasons

Line 199: this is figure 3. But, also line 242 reports figure 3. Check the correspondence between the text and the number of figures and tables.

Line 199: In all the figures homogenize the decimals.

Line 199: I suggest to invert the color ramp of the maps from blue to red (from low to high values). It is more readable.

Line 199: Use in the map and in all the text “°C” and not “C°”.

Line 200: I think it is clearer to use “lake bathymetry” than “spatial lake depth”

Line 200: “and” instead of “&”

Lines 202-203: The sentence is not clear. Please reformulate.

Line 205: resuspension by what? Wind? Fishes?

Lines 206-208: this sentence is not clear. Please rewrite. What do you mean with “flushing rate of the lake”? The residence time is referred to the water residence time of the whole Lake Tana?

Lines 210-211: You did not mentioned water flow values in the different seasons in the previous paragraphs, only the annual MCM. Can be interesting to better quantify this “low flushing rate/high residence time” in the dry season.

Lines 213-214: As the previous sentence. Make more readable and highlight interesting results of the maps.

Line 215: “transporting” with “transport”

Lines 216-218: Please clarify this sentence “that DAP and Urea fertilizer application for rain fed and irrigation agriculture..”. What do you mean? That fertilizers application increased on the soil irrigated for agriculture?

Lines 219-220: Rewrite simplifying the sentence on the temporal variability of the temperature.

Line 221: substitute “presented below” with “presented in Table 1”.

Line 221: The results and  discussion of figure 3e on TDS concentration maps of the three months analysed is missing.

Line 222: Table 1. Standard deviation values should be reported together with mean values. Again, homogenize decimals for the different parameters. Use mg L-1 and not mg/l, and °C and not C°

Lines 223-233: I think that this paragraph is more Material and Methods, and can fill up the gap that I highlighted in the previous paragraph 2.3.2

Line 234: substitute “figure 3” with “figure 4”

Lines 234-235: Please rewrite the sentence. In particular “almost all in all” and delete “measure of acidity or alkalinity”. Leave only pH

Line 240: substitute “described below” with “reported in Table 2”

Line 243: in Table 2 homogenize decimals for TP and TN values. Use °C and not C°

Line 248: homogenize decimals

Line 248-250: This sentence is more material and methods than result

Line 251: substitute “as indicated below” with “as showed in figure 5a”

Line 254: substitute “Figure 4” with “Figure 5”. Add a) and b) next to the figures

Lines 258-259: It seems that this statement is not correct. From figure 5a we can see that in all the three dates the western portion of Lake Tana is suitable for water hyacinth growth

Lines 262-263: “The seasonal surface water fluctuation of the lake [58] affected by the magnitude of infestable area”. This sentence the verb is missing. Please rewrite.

Line 266: correct with “5759 ha of flood plain”

Line 268: substitute “infestation” with “invasion”

Line 270: correct with “deposition from the major river”

Line 271: substitute “below” with “in Table 3”

Line 273: Rewrite the table caption. For example: Summary of the predicted suitable and not-suitable area for the water hyacinth growth.

Line 273: check in the table decimals and thousand values

Line 277: “Fig. 6” instead of “Fig. 5”

Line 278: replace “5 and 6” with “five and six” (up to ten, the numbers should be written in letters)

Line 280: use “littoral portion” instead of “bay part”

Line 290: change “still the area is free” with “the area is still free”

Line 291: substitute “indicated in Figure 4” with “showed in Figure 5”

Line 293: I suggest o use “stagnant water” instead of “quiescent water”

Lines 294-295: It is not possible to obtain wind magnitude and direction data for the Lake Tana basin? These data can improve the discussion.

Line 296: as in the previous comment, use for example “stagnant water and littoral areas”

Line 299: correct with “hyacinth growth in Lake Tana were”

Line 300: are you sure that less than 10% of the total lake can be defined as “substantial”?

Line 300: substitute “infestation” with “invasion”

Line 301: substitute “part” with “portion”

Lines 301-302: I suggest to delete this sentence

Line 302: ass “s” to result”; change the verb in “are”

Line 305: change “The weed needs quiescent water and this type of water is found in the north” with “The weed needs stagnant water, which are typical of the north…”

Round 2

Reviewer 1 Report

I suggest to Authors only a few changes in the text: Row 41 – scientific name species in italics Row 39 – replace "entrance of sun" with “sunlight penetration” Row 40- remove "water hyacinth" and the second bracket because there are too much! Row 44 - ……..has been discovered in 1823 by a German……. Row 47 - reached many tropical (without "to") Rows 380-382 – replace “Ceschin, S., Abati, S., Traversetti, L., Spani, F., Del Grosso, F. and Scalici, M., 2018. Effects of the invasive duckweed Lemna minuta on aquatic animals: evidence from an indoor experiment. Plant Biosystems382 An International Journal Dealing with all Aspects of Plant Biology, pp.1-7” with Ceschin S., Abati S., Traversetti L., Spani F., Del Grosso F., Scalici M., 2019. Effects of the alien duckweed Lemna minuta Kunth on aquatic animals: an indoor experiment. Plant Biosystems. DOI: 10.1080/11263504.2018.1549605

Reviewer 2 Report

The authors have made all the required suggestions. Also, improved the text and the results quite a lot. The discussion is much broader now, and better quality.The format is not equal in the entire manuscript. The title of figures, for instance, are in a different format. The paragraphs as well (e.g. line 54 and 71). figure 3 has very low quality, althoug it is a good explanation for the text. there is no explanation for the TDS in December 2016, the authors MUST provide an explanation. I think something went wrong in the analysis of the TDS. Many commas are missing in the table 3. 
